# Enhancing Prediction Performance by Add-On Combining Circulating Tumor Cell Count, CD45^neg^ EpCAM^neg^ Cell Count on Colorectal Cancer, Advance, and Metastasis

**DOI:** 10.3390/cancers13112521

**Published:** 2021-05-21

**Authors:** Sherry Yueh-Hsia Chiu, Chia-Hsun Hsieh, Jeng-Fu You, Po-Yu Chu, Hsin-Yuan Hung, Pao-Hsien Chu, Min-Hsien Wu

**Affiliations:** 1Department of Health Care Management, College of Management, and Healthy Aging Research Center, Chang Gung University, Taoyuan City 33302, Taiwan; sherrychiu@mail.cgu.edu.tw; 2Division of Hepatogastroenterology, Department of Internal Medicine, Kaohsiung Chang Gung Memorial Hospital, Kaohsiung City 83301, Taiwan; 3Division of Hematology-Oncology, Department of Internal Medicine, New Taipei City Municipal TuCheng Hospital, New Taipei City 23652, Taiwan; wisdom5000@cgmh.org.tw; 4Division of Hematology-Oncology, Department of Internal Medicine, Chang Gung Memorial Hospital at Linkou, Taoyuan City 33302, Taiwan; 5Collage of Medicine, Chang Gung University, Taoyuan City 33302, Taiwan; you3368@cgmh.org.tw (J.-F.Y.); hsinyuan@cgmh.org.tw (H.-Y.H.); pchu@cgmh.org.tw (P.-H.C.); 6Division of Colon and Rectal Surgery, Department of Surgery, Chang Gung Memorial Hospital at Linkou, Taoyuan City 33302, Taiwan; 7Ph.D. Program in Biomedical Engineering, Chang Gung University, Taoyuan City 33302, Taiwan; D0528002@cgu.edu.tw; 8Division of Colon and Rectal Surgery, New Taipei Municipal TuCheng Hospital, New Taipei City 23652, Taiwan; 9The Cardiology Division, Department of Internal Medicine, Chang Gung Memorial Hospital at Linkou, Taoyuan City 33302, Taiwan; 10Graduate Institute of Biomedical Engineering, Chang Gung University, Taoyuan City 33302, Taiwan; 11Department of Chemical Engineering, Ming Chi University of Technology, New Taipei City 24301, Taiwan

**Keywords:** colorectal cancer (CRC), circulating tumor cells (CTCs), carcinoembryonic antigen (CEA), cancer detection, cancer metastasis

## Abstract

**Simple Summary:**

Information describing circulating tumor cells (CTCs) holds promise for clinical applications. However, conventional CTCs enumeration could ignore the CTCs more relevant to cancer metastasis. Thus, negative selection CTC enumeration was proposed, by which information on the numbers of CTCs and CD45^neg^ EpCAM^neg^ cells can be obtained. By combining this approach with the conventional biomarker carcinoembryonic antigen (CEA), this study aimed to explore whether any combination of these biomarkers could improve the predictive performance for colorectal cancer (CRC) or its status. Results revealed that a combination of the two cell populations showed improved performance (AUROC: 0.893) for CRC prediction over the use of only one population. Compared with CEA alone, the combination of the three biomarkers increased the performance (AUROC) for advanced CRC prediction from 0.643 to 0.727. Compared with that of CEA alone for metastatic CRC prediction, the AUROC was increased from 0.780 to 0.837 when the CTC count was included.

**Abstract:**

Conventional circulating tumor cell (CTC) enumeration could ignore the CTCs more relevant to cancer metastasis. Thus, negative selection CTC enumeration was proposed, by which information on two cellular biomarkers (numbers of CTCs and CD45^neg^ EpCAM^neg^ cells) can be obtained. By combining this approach with the conventional biomarker carcinoembryonic antigen (CEA), this study aimed to explore whether any combination of these biomarkers could improve the predictive performance for colorectal cancer (CRC) or its status. In this work, these two cell populations in healthy donors and CRC patients were quantified. Results revealed that enumeration of these two cell populations was able to discriminate healthy donors from CRC patients, even patients with non-advanced CRC. Moreover, the combination of the two cell populations showed improved performance (AUROC: 0.893) for CRC prediction over the use of only one population. Compared with CEA alone, the combination of the three biomarkers increased the performance (AUROC) for advanced CRC prediction from 0.643 to 0.727. Compared with that of CEA alone for metastatic CRC prediction, the AUROC was increased from 0.780 to 0.837 when the CTC count was included. Overall, this study demonstrated that the combination of these two cellular biomarkers with CEA improved the predictive performance for CRC and its status.

## 1. Introduction

Cancer metastasis, the main cause of cancer-related death [1], describes the migration of cancer cells (e.g., via the blood circulation) to areas not directly adjacent to the primary tumor site. In the process of cancer metastasis, cancer cells from the primary tumor that shed into the blood circulation are defined as “circulating tumor cells (CTCs)”; these cells were first documented in 1869 [2]. Reports in the literature have revealed that the existence of CTCs in blood circulation is associated with cancer metastasis [3,4]. Therefore, the information obtained from cancer patients’ CTCs (e.g., their numbers [3,4] or genetic information [3,4]) holds great promise for clinical applications. However, these cells are rare in blood circulation (e.g., 1 CTC per 10^5^~10^7^ blood cells [5]), making them technically challenging to quantify or isolate. Until the first equipment (i.e., CellSearch system) for CTC enumeration was launched in 2014, a report successfully demonstrated the clinical efficacy of CTC enumeration in at least seven different cancer types, including colorectal cancer (CRC) [6]. In that study, briefly, CTCs (i.e., epithelial cells expressing the surface antigen epithelial cell adhesion molecule (EpCAM) in a blood sample) were shown to be extremely rare in healthy subjects and patients with nonmalignant lesions. However, they were detected in larger quantities in numerous metastatic carcinomas [5,6,7].

After that, the concept of identifying and quantifying epithelial cells expressing the surface antigen EpCAM in a blood sample (i.e., normally referred to as positive selection-based CTC detection or enumeration) was commonly adopted in CTC-related studies [5,6,7]. Based on these scientific findings, a wide variety of clinical studies were carried out. Reports in the literature, for example, have demonstrated that cancer patients with a higher number of CTCs normally have a poorer prognosis than those with a lower number of CTCs [8,9,10]. Moreover, the predictive roles of CTCs for specific cancer therapies have also been demonstrated in different scenarios in patients with breast [11], head and neck [12], or prostate cancers [9]. The prognostic role of CTCs has also been proven for patients with CRC after or during anticancer therapy [13,14,15].

Although the conventional CTC enumeration (e.g., CellSearch) approach and relevant assays have been successfully demonstrated for various clinical applications in oncology [5,6,7,8,9,10,11,12,13,14,15], there is an important biological issue that should be considered. As discussed earlier, most of the current CTC isolation or enumeration approaches primarily utilize the cellular surface antigen EpCAM for the identification of CTCs. However, CTCs, particularly those with a metastatic nature, can undergo epithelial-to-mesenchymal transition (EMT) [4,5,6,7,16]. After EMT, the surface expression of EpCAM is significantly downregulated [4,5,6,7,16]. Due to the heterogeneous nature of CTCs, clinically meaningful CTCs closely relevant to cancer metastasis or cancer progression might be missed if conventional positive selection-based CTC isolation methodologies (e.g., CellSearch) are used [5,6,7,16]. Moreover, compared to the above-mentioned biochemical CTC isolation method, the physical methods for CTC isolation (e.g., mainly size-dependent filtration) are normally easy-to-operate, and label-free, but these methods may compromise the viability and purity of isolated CTCs due to the high shear stress condition, cell blockage, and leukocyte contamination during the cell isolation process [16]. This could in turn affect the performance of CTC enumeration. To address this issue, negative selection-based strategies for CTC isolation or enumeration were proposed in recent years [5,6,16,17,18]; in these strategies, only blood cells (i.e., erythrocytes and leukocytes) are depleted using immunomagnetic bead-based cell isolation methods. This approach could thus leave all possible CTCs in the population of remaining cells (i.e., including CD45^neg^ EpCAM^pos^ cells (i.e., the conventionally defined CTCs) and the CD45^neg^ EpCAM^neg^ cell population that could contain the clinically-important CTCs that have undergone EMT). This cell population could be rapidly distinguished, classified, enumerated, and purified via further immunostaining and single-cell analysis techniques (e.g., the flow cytometry used in this study). The quantity of the latter cell population was proven to be significantly increased in the blood samples of cancer patients when compared to those of healthy controls in our previous study [17]. Moreover, high-purity isolation of the CD45^neg^ EpCAM^neg^ cell population from the blood samples of cancer patients was successfully achieved in our previous study [18]. The initial analytical results showed that the CD45^neg^ EpCAM^neg^ cell population in the blood samples of cancer patients might contain cancer-related cells, particularly EMT-transformed CTCs [18]. Taken together, the above findings suggest that the information obtained from the CD45^neg^ EpCAM^neg^ cells (e.g., the cell count) might be clinically meaningful for the prediction of cancer or its disease status.

On the other hand, it is well-known that serum biomolecular markers are clinically utilized to diagnose or monitor CRC onset as well as tumor progression after standard therapies. Among these markers, serum carcinoembryonic antigen (CEA) is the most widely used tumor biomarker in CRC. The role of CEA has been validated in screening [19], surveillance [20], prognosis [21], and decision-making processes for adjuvant therapies in CRC patients [22]. However, the disappointing observation is the low sensitivity and specificity of CEA in CRC [23,24,25]. For example, a report concluded that the sensitivity and specificity of CEA for the detection of CRC recurrence were only 59% and 84%, respectively [23]. One plausible way to improve this sensitivity and specificity is to combine the biomarker CEA with other promising cellular or molecular biomarkers to increase the performance of this approach for CRC diagnosis [24,25].

Based on the above discussions, overall, one question is raised: can the use of any combination of these cellular or molecular biomarkers (i.e., the CD45^neg^ EpCAM^pos^ CTC count, CD45^neg^ EpCAM^neg^ cell count, or CEA level) improve the predictive performance for CRC disease (e.g., earlier diagnosis) or its status (e.g., advanced or metastatic CRC)? To answer this question, a prospective translational trial was carried out in which a negative selection-based CTC enumeration technique [26] was utilized to quantify the two cell populations as described previously in the blood samples of healthy blood donors and CRC patients with different stages of the disease. Therefore, our study aimed to combine CD45^neg^ EpCAM^pos^ CTCs and CD45^neg^ EpCAM^neg^ cells to predict CRC, besides, based on existing conventional serum biomarker CEA, to evaluate add-on combining both CTC molecular biomarkers whether can improve the performance of AUROC for advanced CRC and CRC metastasis prediction.

## 2. Materials and Methods

### 2.1. Enrollment of Healthy Volunteers and CRC Patients

The study was performed under the framework of the Institutional Review Board (IRB) of Chang Gung Memorial Hospital at Linkou, Taiwan (approval ID: 201800501B0 and 201900267B0). Informed consent was obtained from all blood donors, and all methods were carried out in accordance with relevant guidelines. Healthy blood donors were recruited if they (1) had not been diagnosed with cancer by physicians in the past and (2) were older than 20 years. CRC patients were recruited if they (1) had newly diagnosed CRC and had not undergone any cancer treatment, (2) had histologically or cytopathologically confirmed colorectal adenocarcinoma by a pathologist, (3) had disease staging as defined by the 8th edition of the American Joint Committee on Cancer criteria, and (4) were older than 20 years. In this study, CRC patients with synchronous or previous cancer treatment within the last five years were excluded.

### 2.2. Blood Sample Processing Using the Negative Selection-Based Cell Enrichment Scheme

Before the collection of donor blood samples, the first 3–5 mL of the blood sample was discarded to prevent epithelial cell contamination during the blood drawing process. After that, approximately 8–10 mL of the blood sample obtained from the blood donor was stored in vacutainer tubes with tripotassium ethylenediaminetetraacetic acid (K_3_-EDTA) (B.D. Bioscience, San Diego, CA, USA) at 4 °C. Collection was quickly followed by blood sample processing using the negative selection-based cell enrichment scheme, reported on previously [26], within 24 h so ass to avoid the alteration of the gene and surface protein expression [27]. Briefly, the blood sample was first mixed with an erythrocyte lysis buffer (150 mM NH_4_Cl, 3.5 mM KHCO_3_, and 50 μM EDTA (pH 7.0)) and was then incubated for 8 min. After washing with phosphate-buffered saline (PBS), containing 2 mM EDTA, the cell sample obtained was then processed using a commercially available kit (EasySep Human CD45 Depletion Kit, StemCell Technologies, Vancouver, BC, Canada) to deplete leukocytes. All procedures were based on the instructions provided by the manufacturer. Briefly, the cell samples were resuspended in PBS to a final cell density of 1 × 10^8^ cells/mL. The depletion cocktail was then added to the prepared cell suspension. This was followed by 5 min of incubation at room temperature. After that, the well-mixed nanoparticle suspension was added to the treated sample. After 3 min of incubation, the total volume of the sample-nanoparticle mixture was adjusted to 2.5 mL by adding PBS. This was followed by placing the sample-loaded tube into an EasySep™ Magnet (StemCell Technologies, Vancouver, BC, Canada) for 5 min. After that, the cell suspension was aspirated using a pipette without disturbing the magnetically-attracted leukocytes on the tube wall. After the residual buffer was removed by centrifugation, the harvested cell sample was prepared for subsequent cell counting using immunofluorescent dye staining and flow cytometry.

### 2.3. Enumeration of CD45^neg^ EpCAM^pos^ CTCs and CD45^neg^ EpCAM^neg^ Cells by Immunofluorescent Dye Staining and Flow Cytometry

The enriched cell sample was stained with Hoechst 33,342 (6.5 ng/mL; Invitrogen, Thermo Fisher Scientific, Inc., Waltham, MA, USA), an Alexa Fluor 488-conjugated donkey anti-mouse IgG secondary antibody (1000× dilution; Invitrogen, Thermo Fisher Scientific, Inc., MA, USA), and APC-conjugated rabbit anti-human EpCAM antibody (200× dilution; Sino Biological Inc., Chesterbrook, PA, USA). In this study, Alexa Fluor 488-conjugated donkey anti-mouse IgG secondary antibody was used to label the human CD45 marker, which could bind to the residual leukocytes coated with the commercial mouse anti-human CD45 antibodies (EasySep™ Human CD45 Depletion Cocktail II contained in EasySep Human CD45 Depletion Kit, StemCell Technologies, Vancouver, BC, Canada) after the CD45 depletion process. After 1 h of incubation and subsequent washing using PBS, the prepared cell samples were analyzed using flow cytometry (CytoFLEX, Beckman Coulter, CA, USA) to quantify the CD45^neg^ EpCAM^pos^ CTCs and CD45^neg^ EpCAM^neg^ cells.

### 2.4. Statistical Analysis

For dichotomous variables, the proportion was determined, and the $x^{2}$ or Fisher exact test was adopted for statistical checking. For continuous variables, the Anderson–Darling test was used to test the normality, and the Mann–Whitney U test was then adopted to compare the values between the CRC-free and CRC groups if the variables violated a normal distribution. The Kruskal–Wallis test was applied for comparisons among more than two groups (i.e., CRC-free, non-advanced CRC, and advanced CRC). The proportional distribution of patients in binary groups (i.e., advanced vs. non-advanced CRC) or among multiple groups (i.e., stage) was examined using the $x^{2}$ or Fisher exact test. The trend test was also applied for proportion by gradient distribution checking. Multiple logistic regression with adjustment for age and sex were conducted to examine the performance of using CD45^neg^ EpCAM^neg^ CTCs and CD45^neg^ EpCAM^pos^ cell counts to discriminate dichotomous variables as dependent variables, such as CRC-free vs. CRC, non-advanced vs. advanced CRC, and nonmetastatic vs. metastatic CRC. The adjusted odds ratio (adj. OR) with a 95% confidence interval (95%CI) demonstrated the probability of a patient having CRC, advanced CRC, or metastatic CRC. The predictive performance of eAUROC with a 95%CI simultaneously considered sensitivity and specificity. The result was evaluated to assess and compare the performance of prediction models based on the different multiple logistic regression models. The Youden index (sensitivity + specificity − 1) evaluates the optimal cutoff points of CD45^neg^ EpCAM^neg^ cells and CD45^neg^ EpCAM^pos^ CTCs to discriminate between CRC, advanced CRC, and metastatic CRC. All statistical analyses were performed with SAS software version 9.4, and the statistical significance level was defined as a *p*-value less than 0.05.

## 3. Results

### 3.1. Background of the Present Study

In this study, we recruited 85 CRC patients and 73 CRC-free subjects but excluded CRC patients without primary CRC or with missing information on the biomarkers discussed. Therefore, 73 CRC patients and 71 CRC-free subjects were included for analysis (Figure 1).

The age in the CRC group (mean = 62.70) was significantly higher than that in the CRC-free group (mean = 43.38) (Table 1). However, there was no significant difference in the distribution of sex (Table 1). Based on TNM staging information, moreover, 27 (27%) and 46 (63%) patients were classified into the non-advanced (i.e., stages 0–II) and advanced (i.e., stagess III–IV) CRC groups, respectively (Table 1).

Furthermore, there were 22 (30.1%) patients with CEA > 5 ng/mL at diagnosis (Table 1). Regarding the cell biomarker distribution between the CRC and CRC-free groups, as a whole, the counts of CD45^neg^ EpCAM^pos^ CTCs (i.e., conventionally defined CTCs) and CD45^neg^ EpCAM^neg^ cells were statistically higher in the CRC group than the CRC-free group (*p* < 0.0001) (Figure 2). Using the two or more counts as grouping criterion for CD45^neg^ EpCAM^pos^ CTCs, a higher proportion (50.7%) of ≥2 counts in the CRC group was found in comparison with that in the CRC-free group (25.4%) (*p*-value = 0.0018) (Table 1).

### 3.2. Performance of CD45^neg^ EpCAM^pos^ CTCs and CD45^neg^ EpCAM^neg^ Cell Counts for CRC Prediction

There were 144 subjects (71 CRC-free subjects vs. 73 CRC patients) considered to evaluate the performance of CRC prediction. Using the CRC and CRC-free groups as dependent variables for the prediction mode, the logistic regression was employed to estimate the odds ratio (OR) for CTCs and performance of prediction for CTCs (continuous or binary variable) with different cutoffs. For the use of CD45^neg^ EpCAM^pos^ CTC counts for predicting CRC, after adjusting for age and sex, the adjusted odds ratio (adj. OR) was significantly increased 1.50-fold (95%CI: 1.12, 2.00) for each one-count increase in CD45^neg^ EpCAM^pos^ CTCs. For the binary variable using 3 counts as cutoff for CD45^neg^ EpCAM^pos^ CTCs, compared with <3 counts, the adj. OR was 6.10(1.77, 21.06) and AUROC was 0.875. (Table 2).

Regarding the use of the CD45^neg^ EpCAM^neg^ cell counts for predicting CRC, after adjusting for age and sex, the adj. ORs were estimated to be 2.44 (95%CI: 1.06, 5.61), 2.39 (95%CI: 1.02, 5.61), 3.82 (95%CI: 1.54, 9.49), 3.12 (95%CI: 1.22, 7.94), and 2.92 time (95%CI: 1.12, 7.62) for the continuous cell count model for cell count cutoff points of ≥300, ≥400, ≥500, and ≥600, respectively. Taking the sensitivity and specificity into account, the AUROC was 0.867–0.873, but the AUROC calculated using a cutoff point of ≥400 (AUROC = 0.873 (95%CI: 0.815, 0.931)), was higher than those calculated with the other approaches (Table 2). Based on the above evaluations, we further combined the two parameters of CD45^neg^ EpCAM^pos^ CTCs (i.e., the continuous cell count model) and CD45^neg^ EpCAM^neg^ cells (i.e., the cutoff point of ≥400 cells) to predict the presence or absence of CRC. The adj. ORs were calculated to be 1.42 (1.07, 1.89) and 2.84 (95%CI: 1.10, 7.35), respectively, after adjustment for age and sex. The AUROC was 0.893 (95%CI: 0.842, 0.944) (Table 2 and Figure 3). This AUROC (i.e., based on the combination of both cell biomarkers) was revealed to have better performance for predicting CRC than any one of these two cell biomarkers alone.

### 3.3. Performance Evaluation of Using the CD45^neg^ EpCAM^pos^ CTCs and CD45^neg^ EpCAM^neg^ Cell Counts for the Prediction of Advanced or Metastatic CRC

Regarding the distribution of CD45^neg^ EpCAM^pos^ CTC counts in 7 mL blood samples, the median counts were experimentally determined to be 0, 2.0, and 1.1 for the CRC-free, non-advanced CRC, and advanced CRC groups, respectively. For the two status comparison, we conducted the Mann–Whitney U test for every comparison of two groups. The results showed significant differences between the CRC-free vs. non-advanced CRC groups (*p* < 0.0001) and the CRC-free vs. advanced CRC groups (*p* = 0.0049) (Figure 4A). Moreover, for nonmetastatic and metastatic CRC, the median counts were 0, 2.0, and 1.0 for the CRC-free, nonmetastatic CRC, and metastatic CRC groups, respectively. The results also demonstrated a significant difference between the CRC-free vs. nonmetastatic CRC groups (*p* < 0.0001) and the CRC-free vs. metastatic CRC groups (*p* = 0.0305) (Figure 4B). Regarding the distribution of CD45^neg^ EpCAM^neg^ cells in 7 mL blood samples, the median counts were 193, 332, and 461.9 for the CRC-free, non-advanced CRC, and advanced CRC groups, respectively. On the other hand, the median counts were 193, 396, and 425.8 for the CRC-free, nonmetastatic CRC, and metastatic CRC groups, respectively. We noted significant differences between the CRC-free vs. non-advanced CRC groups (*p* = 0.0018) and the CRC-free vs. advanced CRC groups (*p* = 0.0002) (Figure 4C). The results also exhibited similar patterns regarding metastatic status (Figure 4D). Our results demonstrated a significant difference between nonmetastatic/metastatic or non-advanced/advanced CRCs compared with CRC-free, regardless of the tumor status.

### 3.4. Discrimination of Cancer Stages and Advanced/Metastatic CRC by Combining the Biomarkers of the CD45^neg^ EpCAM^pos^ CTCs, CD45^neg^ EpCAM^neg^ Cell Count, and CEA Level

There were 73 CRC patients only were included for advanced/metastatic CRC prediction evaluation (non-advanced: advanced 27:46; non-metastatic: metastatic 59:14). To investigate whether the counts of CD45^neg^ EpCAM^neg^ cells or CD45^neg^ EpCAM^pos^ CTCs can improve the performance of the model for discriminating CRC at different stages or with different statuses (i.e., advanced CRC or metastatic CRC), the proportion distributions with the different approaches were assessed and conducted the chi-square test for distribution examination. When one biomarker alone (i.e., CD45^neg^ EpCAM^neg^ cell count, CD45^neg^ EpCAM^pos^ CTCs, or CEA level alone) was used for discrimination, first, neither CD45^neg^ EpCAM^pos^ CTCs (≥3/7 mL as a cutoff) nor CD45^neg^ EpCAM^neg^ cells (≥500/7 mL as a cutoff) exhibited significant proportion differences (Appendix A). However, the conventional CEA level with >5 ng/mL as a cutoff point significantly discriminated disease stage (*p* = 0.0078), advanced CRC (*p* = 0.0288), and metastatic CRC (*p* = 0.0020) (Appendix A). Furthermore, we combined all possible pairs of two of these three biomarkers (i.e., CD45^neg^ EpCAM^pos^ CTC count (≥3/7 mL as a cutoff), CD45^neg^ EpCAM^neg^ cell count (≥500/7 mL as a cutoff), and CEA level (>5 ng/mL as a cutoff)) to distinguish CRC stages and statuses. The results (Appendix A) revealed that the combination of the CD45^neg^ EpCAM^neg^ cell count and/or CD45^neg^ EpCAM^pos^ CTC count was not able to significantly discriminate the stage and status (i.e., metastatic or advanced) of CRC. Conversely, the combination of either the CD45^neg^ EpCAM^neg^ cell count or CD45^neg^ EpCAM^pos^ CTCs with the CEA level significantly discriminated the stage and advanced/metastatic status of CRC (Appendix A). A combination of the three explored biomarkers, furthermore, showed that this approach was able to significantly discriminate the CRC stage (*p* = 0.0352) and differentiate between advanced vs. non-advanced CRC (*p* = 0.0282) and metastatic vs. nonmetastatic CRC (*p* = 0.0088) (Figure 5).

Based on the above findings, a series of statistical analyses using different cutoff points and continuous situations, as shown in Appendix A was carried out to determine the optimum cutoff point of each studied biomarker to improve the prediction of advanced or metastatic CRC by using multiple logistic regression for age, gender adjustment. Based on the statistical evaluations, overall, the optimum cutoff points for the biomarkers explored were set at CD45^neg^ EpCAM^pos^ CTCs (≥3/7 mL), CD45^neg^ EpCAM^neg^ cells ≥ 400/7 mL, and CEA > 5 ng/mL for the prediction of advanced or non-advanced CRC. Under these conditions, the AUROC any single biomarker for the prediction of advanced CRC was in the range of 0.614 to 0.643 (Table 3).

The performance of the AUROC (i.e., 0.672 to 0.712) was improved when any two of these biomarkers were combined. Within the experimental conditions investigated, furthermore, the AUROC of all three biomarkers combined was 0.727 (95%CI: 0.0609, 0.845) for the prediction of non-advanced vs. advanced CRC (Table 3). Compared with the conventional biomarker CEA for clinical use, the combination of these three explored biomarkers can improve the predictive performance (AUROC) for advanced CRC from 0.643 to 0.727 (Table 3 and Figure 6).

For the prediction of metastatic vs. nonmetastatic CRC, the optimum cutoff points for the explored biomarkers were set at CD45^neg^ EpCAM^pos^ CTCs (continuous), CD45^neg^ EpCAM^neg^ cells ≥ 400/7 mL and CEA > 5 ng/mL based on the data in Appendix A. For the prediction of metastatic CRC, the AUROC based on the combination of the CD45^neg^ EpCAM^pos^ CTCs (continuous) and CEA level was evaluated to be 0.837 (95%CI: 0.740, 0.934), which was the same as the AUROC (0.837 (95%CI: 0.739, 0.935)) based on the combination of the three investigated biomarkers (Table 4).

Based on the two combinations mentioned above, overall, the predictive performance (AUROC) for metastatic CRC was higher than that for the other conditions within the tested experimental conditions. Compared with the conventional use of the CEA level alone for the prediction of metastatic CRC, the AUROC was increased from 0.780 to 0.837 when the CD45^neg^ EpCAM^pos^ CTC count was added (Figure 7).

## 4. Discussion

Based on the proposed negative selection-based protocol for cell enumeration [26], the concentration of CD45^neg^ EpCAM^pos^ CTCs and CD45^neg^ EpCAM^neg^ cells in a blood sample was proven to be able to significantly discriminate between healthy donors and patients with any stage of colorectal cancer (Figure 2), even though the pathological stage of many enrolled CRC patients was early (stages 0–II) (Figure 4A,C). This method of CTC counting has the limitations of (1) background cell contamination when CTC is extremely rare; (2) incomplete or oversaturated antibody staining resulting in underestimated or overestimated cell counting. Fortunately, these issues could be mostly prevented by non-staining, isotype negative controls, and positive controls using cancer cell lines along with each CTC testing. In addition, it was found that the numbers of CTCs in the advanced CRC (stages III–IV with lymph node involvement or distant metastasis) and non-advanced CRC groups (stages 0–II, without lymph node involvement or distant metastasis) were different. The CTC count was higher in the non-advanced CRC group than in the advanced CRC group, although there was no statistical significance between the groups (Figure 4A). Conversely, the CD45^neg^ EpCAM^neg^ cell count was higher in the advanced CRC group than in the non-advanced CRC group (Figure 4C). The possible explanations for these phenomena might be due to the dissemination of CD45^neg^ EpCAM^pos^ CTCs from the primary tumor during the early stage of cancer and the possible occurrence of EMT at the late stage of cancer, as also described in the introduction section [4,5,6,7,16]. Owing to the likelihood of EMT in cancer disease, adding information on the CD45^neg^ EpCAM^neg^ cell counts to information based on the conventional CD45^neg^ EpCAM^pos^ CTCs might produce a more adequate biomarker combination for CRC prediction. This possibility was also preliminarily demonstrated in this study, showing that the utilization of the counts of both CD45^neg^ EpCAM^pos^ CTCs and CD45^neg^ EpCAM^neg^ cells in blood samples resulted in better performance for CRC prediction than utilizing the conventional CTC count alone (Table 2). Considering the high clinical application potential of CD45^neg^ EpCAM^neg^ cells shown in this study, moreover, the details of this cell population (e.g., their physiological role) are suggested to be subjects of further investigation.

Moreover, as one of the important tumor biomarkers, CEA has been used for the clinical evaluation of cancer patients for more than 30–40 years. As described in the introduction, however, the low sensitivity and specificity of CEA in CRC patients might mislead physicians when making medical decisions. In this study, the results revealed that only 30.1% of CRC patients had positive CEA detection results (Table 1). Similar results were also reported in other literature [23,24,25,28,29]. In 2012, Su et al. reported the low sensitivity (37.0%) of CEA for primary CRC based on 413 CRC cases using the same CEA cutoff point (>5 ng/mL) used our study [28]. In 2015, moreover, Sørensen et al. conducted a systematic review based on 42 original studies to investigate the performance of CEA for the prediction of CRC recurrence. These results also showed that the sensitivity of CEA was approximately 50–80%, which is not effective for detecting CRC recurrence at an early stage [29]. Therefore, in 2019, Marcuello et al. provided a comprehensive review and emphasized future applications for new noninvasive biomarkers, e.g., CTCs and circulating tumor DNA (ctDNA), for CRC screening and management [30]. In this study, the results revealed that the combination of conventional CEA values with CTC counts (i.e., number of CD45^neg^ EpCAM^pos^ CTCs or CD45^neg^ EpCAM^neg^ cells) can significantly discriminate CRC’s advanced or metastatic status (Figure 5, Figure 6 and Figure 7 and Table 3 and Table 4). These findings are also similar to those of other studies [31,32,33]. First, Aggarwal et al. observed that the performance of cancer prognosis among metastatic CRC patients can be distinguished by a combination of the CEA level and baseline CTC count with a cutoff point of 3/per 7.5 mL, suggesting that the performance of the CEA level could be improved by combining it with the CTC marker [31]. Zheng et al. noticed that the combination of the CTC count with the CEA level could be a tool with high diagnostic efficacy for early lung cancer diagnosis [32]. Shi et al. proved that CTCs expressing MAGE3 (melanoma-associated antigen 3), survivin, and CEA were predictive of cryosurgery’s efficacy in unresectable hepatocellular carcinoma [33]. Taken together, these findings supported the use of more than one kind of biomarker (e.g., CD45^neg^ EpCAM^pos^ CTCs, CD45^neg^ EpCAM^neg^ cells, or CEA) by clinicians for the prediction of cancer or its status.

The majority of CRC patients come from sporadic cases, around 80%, others are partially derived from family history and hereditary susceptibility interacting with environmental factors. The CRC incidence rate is highly correlated with age; for those who tend to have a high risk of sporadic CRCs, they have a high likelihood of being late-onset cases, but with a high probability of having MSI (microsatellite instability), MLH1 methylated, BRAF, and KARS mutation [34]. Some studies also demonstrated the mutation associated with advanced and metastasis CRCs [35]. Based on the meta-analysis evidence, these biomarkers also play important roles for the clinical treatment of targeted therapy designed by epidermal growth factor receptor (EGFR) signaling [36]. Therefore, biomarker examination is emerging from biotech development, especially the isolation methods from blood samples [37,38] rather than from tumor tissues, which is convenient for clinical practice using CTCs detection [39]. Based on the development of CTCs examination and knowledge about the specific biomarkers for CRC targeted therapy application, it is promising to combine these together, aiming toward precision medicine and healthcare applications for CRC prognosis prediction. Further add-on molecular markers, such as circulating DNA or mutation of RAS-BRAF genes on CTCs, are highly warranted. Recently, Toh et al. reported those CRC patients with a high level of MSI g CTCs tend to increase CTCs. Furthermore, comparing the different time-points with pre-, intra-, post-operation for CRC, those CRCs with high MSI were significantly associated with increasing CTCs level during intra- and post-operative time points [40], which indicated the CTCs might be influenced by clinical intervention, i.e., chemotherapy, surgery, or radiotherapy, etc. Therefore, the multiple time points with repeated measures for CTCs with longitudinal follow-up would need for future application.

Considering the high predictive performance for CRC and its status (i.e., AUROC of 0.893 for CRC prediction, 0.727 for advanced CRC prediction, and 0.837 for metastatic CRC prediction) achieved by combining multiple biomarkers in this study, the presented method could play a role in the current CRC prevention policy. Taking Taiwan as an example, a nationwide CRC screening program using fecal immunochemical testing (FIT) was launched in 2004, and the effectiveness of this program based on results from 2004–2009 results with a 21.4% screening rate showed a 10% reduction in CRC mortality after adjustment for self-selection bias. Using the simulation approach, the effectiveness could be increased to 36% if the screening rate was increased to 60% [41]. According to the results of an investigation by Andreas et al. in 2014, which investigated patients’ willingness to participate in noninvasive CRC screening, such as stool- or blood-based screening, 83% and 15% of patients were willing to participate and select blood and stool tests, respectively [42]. Obviously, patients tend to undergo blood tests for CRC screening. In addition to the logistical arrangement for nationwide programs, encouraging refusers to participate in screening is currently very challenging in many countries. Taking willingness and accessibility into account and considering the results of our study, these CTC biomarkers might provide promising alternative methods for screening those target populations. On the other hand, however, the colonoscopy referral rate in Taiwan, 71%, lags behind that in Western countries (>90%). Based on these data, FIT-positive cases were followed up until the end of 2012, and a significantly increased risk of CRC-related death was noted, with a 1.64-fold increase in colonoscopy noncompliers compared with compliers [43]. This finding indicates that the effectiveness of CRC screening programs using FIT would be reduced by colonoscopy noncompliers with positive FIT, but this phenomenon can be improved using additional biomarkers, for example, CTCs, which can be expected to first help CRC prediction and then convince patients to undergo a colonoscopy for confirmation and reduce the number of missing patients who have CRC but refuse colonoscopy.

## 5. Conclusions

Compared with the conventional CTC enumeration scheme (e.g., CellSearch), negative selection-based CTC enumeration was found to obtain all possible CTCs [i.e., CD45^neg^ EpCAM^pos^ cells (i.e., conventionally defined CTCs) and the CD45^neg^ EpCAM^neg^ cell population that could contain clinically important CTCs more relevant to cancer metastasis) in blood samples of cancer patients. By combining this approach with an assessment of the conventional serum molecular biomarker CEA, this study aimed to investigate whether any combination of these three biomarkers (i.e., the CD45^neg^ EpCAM^pos^ CTC count, CD45^neg^ EpCAM^neg^ cell count, or CEA level) could improve the predictive performance for CRC or its status. To this end, a clinical trial was carried out in which negative selection-based CTC enumeration was utilized to quantify these two cell populations in the blood samples of healthy donors and CRC patients with different stages of the disease. The results revealed that the number of CD45^neg^ EpCAM^pos^ CTCs or CD45^neg^ EpCAM^neg^ cells was proven to be able to significantly discriminate the healthy donors from the CRC patients, even though the cancer patients had non-advanced CRC (i.e., stages 0–II). In addition, the combinations of CD45^neg^ EpCAM^pos^ CTCs and CD45^neg^ EpCAM^neg^ cells showed improved performance (AUROC: 0.893) for CRC prediction compared with that of either cell population alone. Moreover, compared with conventional biomarker CEA alone, the combination of the three explored biomarkers improved the performance (AUROC) for advanced CRC prediction from 0.643 to 0.727. Similarly, compared with the CEA level alone, combining CD45^neg^ EpCAM^pos^ CTCs with the CEA level resulted in an increase in the AUROC from 0.780 to 0.837. Overall, this study demonstrated that the combination of the two cellular biomarkers obtained via a negative selection-based CTC enumeration scheme with the conventional serum biomarker CEA improved the predictive performance for CRC and its status.

## Figures and Tables

**Figure 1 cancers-13-02521-f001:**
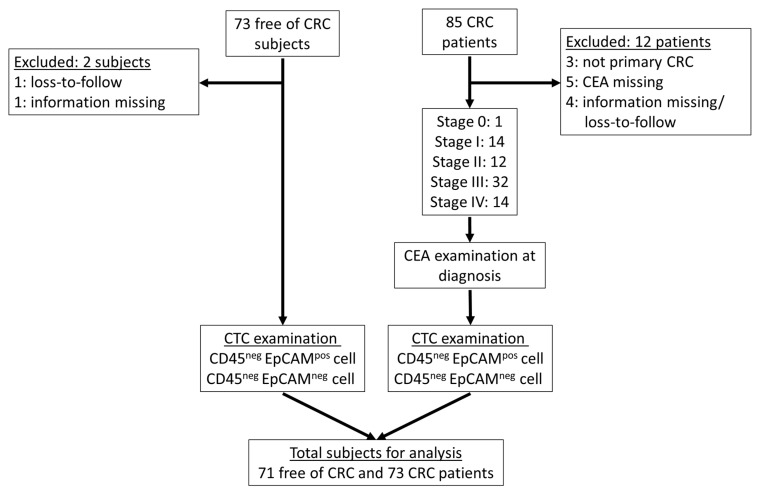
The flowchart of study implementation (after excluding the unqualified cases, 73 CRC patients and 71 CRC-free subjects were included for the following CTC examination).

**Figure 2 cancers-13-02521-f002:**
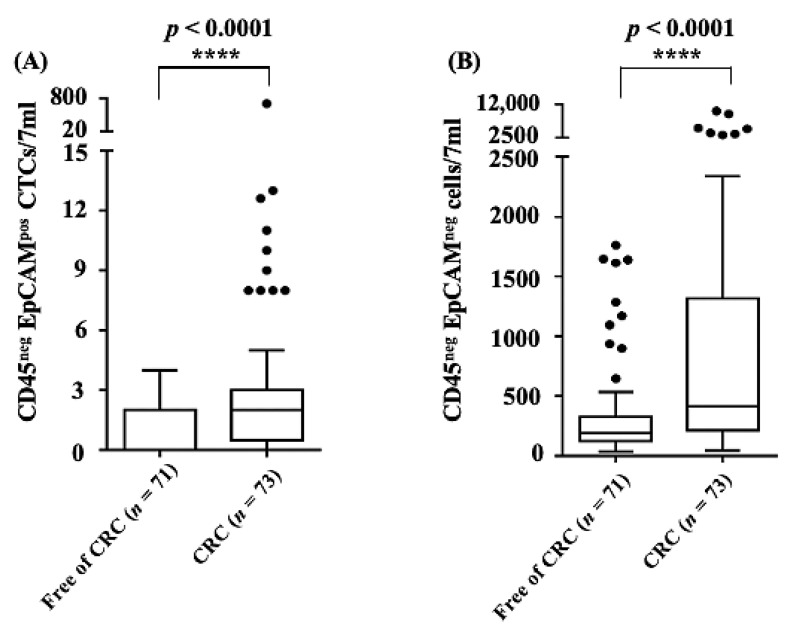
The distribution of (**A**) CD45^neg^ EpCAM^pos^ CTCs and (**B**) CD45^neg^ EpCAM^neg^ cell counts (i.e., cell number/7 mL blood sample) by free of CRC (*n* = 71) and CRC (*n* = 73) status (****: statistical difference *p* < 0.0001).

**Figure 3 cancers-13-02521-f003:**
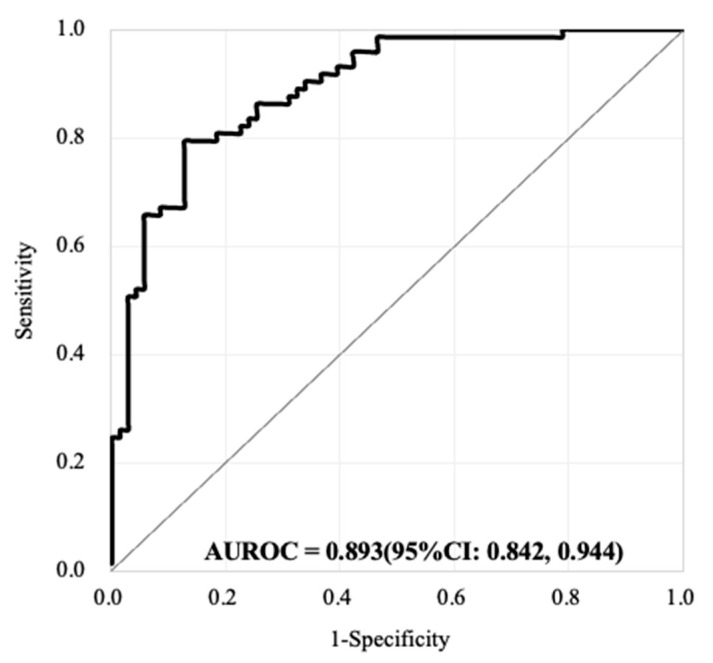
The performance of AUROC combining both CD45^neg^ EpCAM^pos^ CTCs (based on the continuous cell count model) CD45^neg^ EpCAM^neg^ cell counts (based on the cutoff point of ≥400 cells) for the prediction of the presence or absence of CRC (71 free of CRC subjects vs. 73 CRC patients). The AUROC was evaluated to be 0.893 (95%CI: 0.842, 0.944).

**Figure 4 cancers-13-02521-f004:**
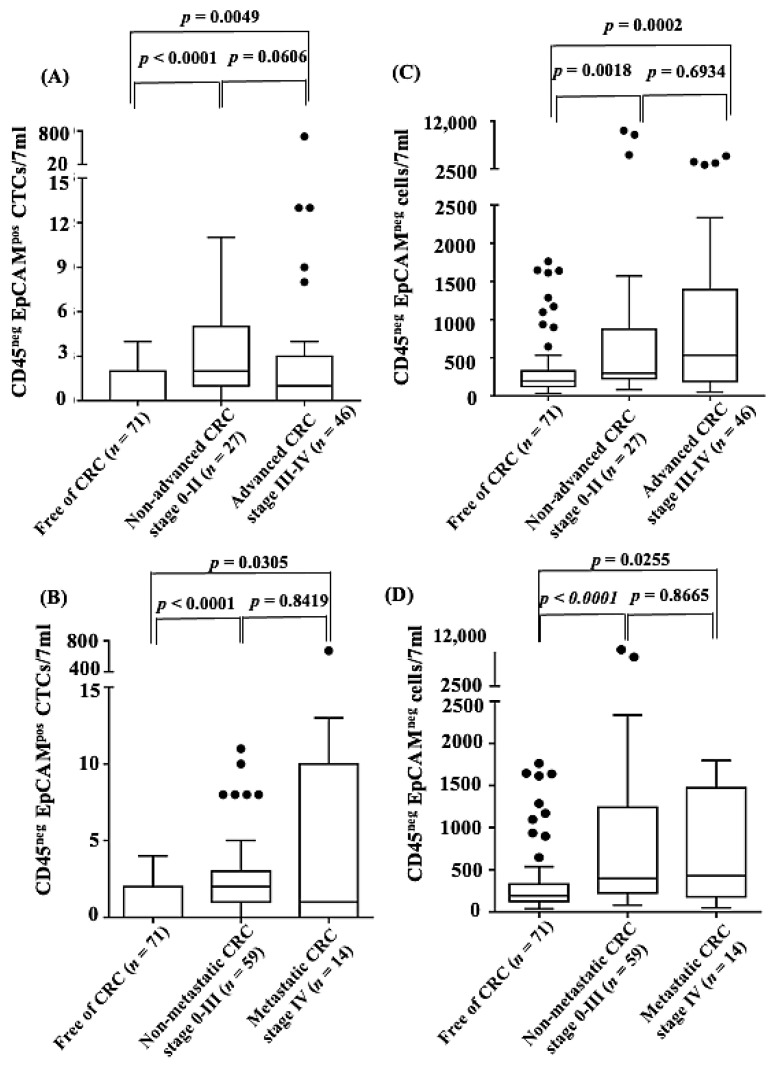
The distribution of CD45^neg^ EpCAM^pos^ CTCs (cell number/7 mL blood sample) by CRC-free (*n* = 71) and (**A**) advanced (*n* = 46)/non-advanced (*n* = 27) CRC status or (**B**) metastatic (*n* = 14)/non-metastatic (*n* = 59) CRC status, and CD45^neg^ EpCAM^neg^ cells (cell number/7 mL blood sample) by CRC-free and (**C**) advanced (*n* = 46)/non-advanced (*n* = 27) CRC status or (**D**) metastatic (*n* = 14)/non-metastatic (*n* = 59) CRC status. (Statistical difference: *p* < 0.05).

**Figure 5 cancers-13-02521-f005:**
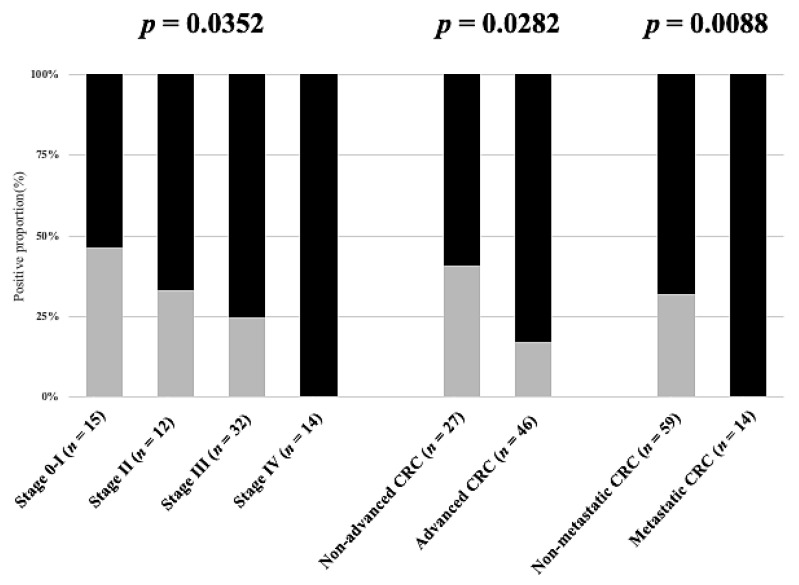
The proportional distribution comparison by advanced or metastatic statuses combing CD45^neg^ EpCAM^pos^ CTCs ≥ 3/7 mL, or/and CD45^neg^ EpCAM^neg^ cell counts ≥ 500/7 mL, or/and CEA > 5 ng/mL. (73 CRC patients only).

**Figure 6 cancers-13-02521-f006:**
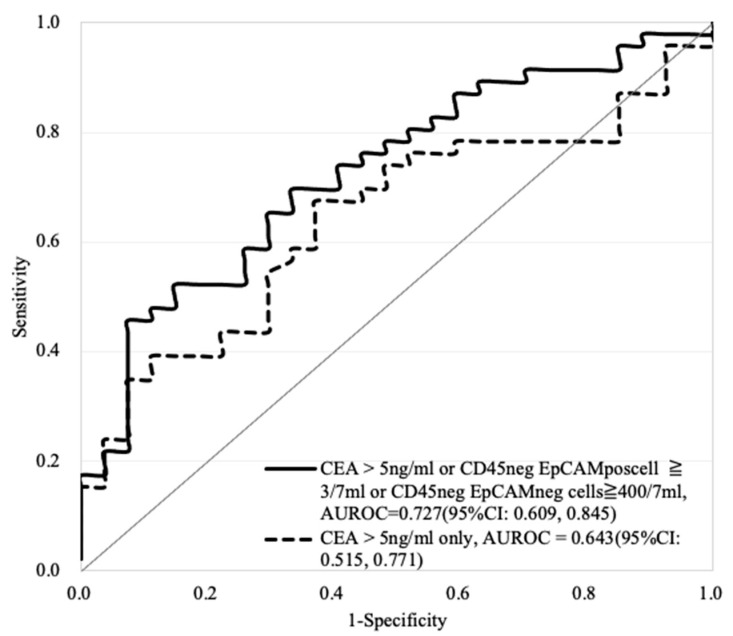
The comparison of AUROC performance based on the use of CEA (>5 ng/mL) only (the evaluated AUROC: 0.643) and the combination of CD45^neg^ EpCAM^pos^ CTCs (≥3 cells/7 mL blood sample) or CD45^neg^ EpCAM^neg^ cell counts (≥400 cells/7 mL blood sample) or CEA (>5 ng/mL) (the evaluated AUROC: 0.727) for advanced CRC prediction. (non-advanced vs. advanced = 27 vs. 46).

**Figure 7 cancers-13-02521-f007:**
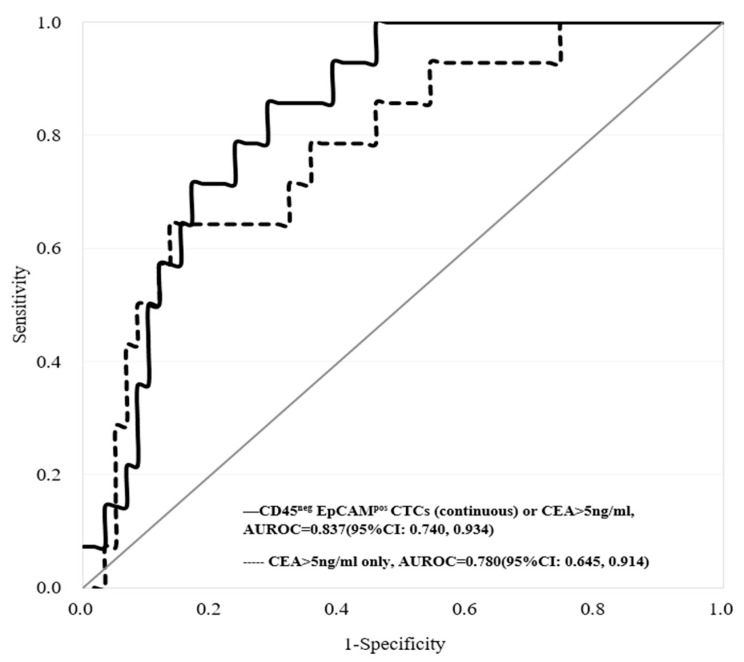
The comparison of AUROC performance based on the use of CEA (>5 ng/mL) only (the evaluated AUROC: 0.780) and combining both CD45^neg^ EpCAM^pos^ CTCs (based on the continuous cell count model) or CEA (>5 ng/mL) (the evaluated AUROC: 0.837) for metastasis CRC prediction. (non-metastatic vs. metastatic = 59 vs. 14).

**Table 1 cancers-13-02521-t001:** The characteristics of study subjects.

Variable	Classification	Free of CRC		CRC		*p*-Value
		No./Mean	%/±SD	No./Mean	%/±SD	
Overall		71		73		
Gender	Female	39	54.9%	31	42.5%	0.1346
	Male	32	45.1%	42	57.5%	
Mean age		43.38	±12.17	62.70	±13.06	<0.0001
Age group	Age < 65 y/o	66	93.0%	39	53.4%	<0.0001
	Age ≥ 65 y/o	5	7.0%	34	46.6%	
TNM	Stage 0	-		1	1.4%	-
stage	Stage I	-		14	19.2%	
	Stage II	-		12	16.4%	
	Stage III	-		32	43.8%	
	Stage IV	-		14	19.2%	

CEA level (ng/mL)	≤5	-		51	69.9%	-
	>5	-		22	30.1%	

CD45^neg^ EpCAM^neg^ cells	(×10^3^)	0.3456	±0.4162	1.0958	±1.824	<0.0001
	median	0.1930		0.4120		

CD45^neg^ EpCAM^pos^ CTCs	<2	53	74.6%	36	49.3%	0.0018
	≥2	18	25.4%	37	50.7%	

	<3	66	93.0%	49	67.1%	0.0001
	≥3	5	7.0%	24	32.9%	

	<4	68	95.8%	57	78.1%	0.0017
	≥4	3	4.2%	16	21.9%	

CRC: colorectal cancer; CEA: Carcinoembryonic Antigen.

**Table 2 cancers-13-02521-t002:** Performance of using CD45neg EpCAMpos CTCs and CD45neg EpCAMneg cell count for predicting colorectal cancer **.

Variable	Classification	Adj. OR ^#^ (95%CI) *	*p*-Value	AUROC (95%CI)
CD45^neg^ EpCAM^pos^ CTCs	(counts, continuous)	1.50 (1.12, 2.00)	0.0065	0.882 (0.828, 0.937)
	count ≥2 vs. <2	1.99 (0.84, 4.75)	0.1196	0.864 (0.803, 0.925)
	count ≥3 vs. <3	6.10 (1.77, 21.06)	0.0042	0.875 (0.817, 0.933)
	count ≥4 vs. <4	5.12 (1.19, 22.11)	0.0285	0.865 (0.804, 0.925)
CD45^neg^ EpCAM^neg^ cells	(counts, continuous)	2.44 (1.06, 5.61)	0.0354	0.868 (0.807, 0.928)
	≥300 vs. <300	2.39 (1.02, 5.61)	0.0449	0.867 (0.805, 0.926)
	≥400 vs. <400	3.82 (1.54, 9.49)	0.0039	0.873 (0.815, 0.931)
	≥500 vs. <500	3.12 (1.22, 7.94)	0.0172	0.869 (0.809, 0.929)
	≥600 vs. <600	2.92 (1.12, 7.62)	0.0290	0.868 (0.807, 0.928)
CD45^neg^ EpCAM^pos^ CTCs	(counts, continuous)	1.42 (1.07, 1.89)	0.0164	0.893 (0.842, 0.944)
CD45^neg^ EpCAM^neg^ cell ≥400		2.84 (1.10, 7.35)	0.0313	

* all analysis was adjusted age (continuous) and gender; #: adj. OR: adjusted odds ratio; AUROC: area under the receiver operating characteristic. **: analysis included 71 free of CRC subjects vs. 73 CRC patients.

**Table 3 cancers-13-02521-t003:** Performance of using CEA, CD45^neg^ EpCAM^pos^ CTC count, CD45^neg^ EpCAM^neg^ cell count, or the combination of them for predicting advanced CRCs **.

Biomarker(s)	Predictor(s) *			Advanced vs. Non-Advanced CRC AUROC (95%CI)
	CD45^neg^ EpCAM^pos^ CTCs ≥3/7 mL	CD45^neg^ EpCAM^neg^ Cells ≥400/7 mL	CEA > 5 ng/mL	
1-type	√			0.614 (0.478, 0.749)
		√		0.635 (0.491, 0.779)
			√	0.643 (0.515, 0.771)
2-type	√	√		0.672 (0.542, 0.803)
		√	√	0.712 (0.587, 0.836)
	√		√	0.677 (0.554, 0.800)
3-type	√	√	√	0.727 (0.609, 0.845)

* all analysis was adjusted age (continuous) and gender; AUROC: area under the receiver operating characteristic; CEA: Carcinoembryonic Antigen. **: analysis included 73 CRC patients only (non-advanced vs. advanced = 27 vs. 46).

**Table 4 cancers-13-02521-t004:** Performance of using CEA, CD45^neg^ EpCAM^pos^ CTC count, CD45^neg^ EpCAM^neg^ cell count, or the combination of them for predicting metastasis CRCs **.

Biomarker(s)	Predictor(s) *			Metastasis vs. Non-Metastasis CRC AUROC (95%CI)
	CD45^neg^EpCAM^pos^ CTCs (Continuous)	CD45^neg^ EpCAM^neg^ Cells ≥400/7 mL	CEA > 5 ng/mL	
1-type	√			0.664 (0.497, 0.831)
		√		0.630 (0.475, 0.784)
			√	0.780 (0.645, 0.914)
2-type	√	√		0.662 (0.495, 0.830)
		√	√	0.786 (0.661, 0.911)
	√		√	0.837 (0.740, 0.934)
3-type	√	√	√	0.837 (0.739, 0.935)

* all analysis was adjusted age (continuous) and gender; AUROC: area under the receiver operating characteristic; CEA: Carcinoembryonic Antigen. **: analysis included 73 CRC patients only (non-metastatic vs. metastatic 59:14).

## Data Availability

Data available on request due to restrictions on ethical consideration.

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
