# Peer review of "Enhancing Prediction Performance by Add-On Combining Circulating Tumor Cell Count, CD45neg EpCAMneg Cell Count on Colorectal Cancer, Advance, and Metastasis"

_cancers, 2021, doi:10.3390/cancers13112521_

Round 1

Reviewer 1 Report

The current manuscript by Sherry et. al. highlighted the combination of the two cellular biomarkers obtained via a negative selection-based CTC enumeration scheme with the conventional serum biomarker CEA improved the predictive performance for CRC and its status. Particularly, the combination of the two cell populations showed improved performance (AUROC: 0.893) for CRC prediction over the use of only one population. The methods in the manuscript seem clearly described, and overall, the authors were able to convey a message about their work. However, there are some technical comments from my side.

comments:
-Several places of the manuscript in the Results section, authors lacked to provide details about their analysis, therefore, I would recommend authors to work on it and simply their explanation.
-The title of the manuscript needs rework, just stating their findings would improve it. 
-I am not sure if the journal needs a simple summary before the abstract, plz check the journal formatting guidelines. 
-English language in the manuscript needs to be improved. 
-Figure captions are not self-explanatory, authors should work on them. 
-For Figures 2 and 4, for both panels authors should use the same scaling, to get a better view of comparison between these data.

These comments are just to improve and polish the manuscript before, it may be considered for acceptance. 

Author Response

Reviewer 1

General Comments:

The current manuscript by Sherry et. al. highlighted the combination of the two cellular biomarkers obtained via a negative selection-based CTC enumeration scheme with the conventional serum biomarker CEA improved the predictive performance for CRC and its status. Particularly, the combination of the two cell populations showed improved performance (AUROC: 0.893) for CRC prediction over the use of only one population. The methods in the manuscript seem clearly described, and overall, the authors were able to convey a message about their work. However, there are some technical comments from my side.

These comments are just to improve and polish the manuscript before, it may be considered for acceptance.

Response: The professional English writing, grammar, punctuation, and spelling for our manuscript were edited by highly qualified native English speaking editors. The editorial certificate has also been uploaded with this revision for the journal’s reference (see “Response of Specific comment 4”).

Specific comment 1:

Several places of the manuscript in the Results section, authors lacked to provide details about their analysis, therefore, I would recommend authors to work on it and simply their explanation.

Response: Agreed. We added some paragraphs to describe methods and explain these results. Based on the CRC and free of CRC as dependent variable for prediction mode, the logistic regression was employed to estimate the odds ratio (OR) for CTCs and performance of prediction for CTCs (continuous or binary variable) with different cutoff…... As the binary variable using 3 counts as cutoff for CD45neg EpCAMpos CTCs, compared with <3 counts, the adj. OR was 6.10(1.77, 21.06) and AUROC was 0.875.” We also added two paragraphs on Results 3.3 and others. (Please refer to the manuscript of Highlighted Revision; Page 8, 9, 11).

Specific comment 2:

The title of the manuscript needs rework, just stating their findings would improve it.

Response: Agreed. We amended our title: “Enhancing prediction performance by add-on combining circulating tumor cell count, CD45neg EpCAMneg cell count on colorectal cancer, advance, and metastasis”. (see manuscript title)

(Please refer to the manuscript of Highlighted Revision; Page 1).

Specific comment 3:

I am not sure if the journal needs a simple summary before the abstract, plz check the journal formatting guidelines.

Response: Thanks for reminding. Yes, besides Abstract, in light of instructions for author, we have added one “Graphical Abstract” to summarize our manuscript (see Graphical Abstract, below).

 Please see picture in attached file

Specific comment 4:

English language in the manuscript needs to be improved.

Response: Please see the first response. We responded- “The professional English writing, grammar, punctuation, and spelling for our manuscript were edited by highly qualified native English speaking editors. The editorial certificate (below) has also been uploaded with this revision for the journal’s reference.”

Please see picture in attached file

Specific comment 5:

Figure captions are not self-explanatory, authors should work on them.

Response: Reviewer’s suggestion is appreciated. The figure captions in this article have now been improved.

(Please refer to the manuscript of Highlighted Revision; Page 6-7, 9-11, 13-14).

Specific comment 6:

For Figures 2 and 4, for both panels authors should use the same scaling, to get a better view of comparison between these data.

Response: No, both were examined on different types, CTC counts and CD45neg EpCAMneg cells, based on 7ml blood. Therefore, the scales for both are different.

13).

Reviewer 2 Report

Revision Manuscript cancers-1169819

The manuscript deals with a hot-topic in the CTC field: EpCam positive CTCs versus EpCaM negatives ones.

The main problem in distinguishing these two population is analytical and derives from the sensitivity and the specificity of the used approach.

The proposed method uses just one EpCam antibody for positive selection, but does not use a CD45 fluorescently label antibody to verify the population obtained after depletion. It seems that the CD45neg/EpCaM neg cells are really very numerous to be all CTCs.

The authors should comment on this and compare their results with studies not using the recognition of biological markers for CTC selection like those base on the physic properties of the CTCs (filtration, dielectrophoresis, or other principles). Moreover, the comparison with the control population requires a matched group for age in which the values of CEA have been measured (CEA varies with the subject's age).

Although the question the authors aim to solve is interesting, the method used to achieve the goal should be carefully evaluated and correctly compared to an appropriate control group. Below you will find some of the major concerns relatively to the submitted manuscript.

Major comments:

  • The authors often refer to “Conventional CTC enumeration” apparently referring to CellSearch CTC counting. In my opinion would be better to specify the method they refer to, since there are other methods for CTC counting (e.g. those based on filtration) that don’t show this drawback.
  • Page 2 line 70 and line 75: the authors refer to one report published 2014 (I guess the one from Cristofanilli), but they added 3 different, more recent references. Please modify references and/or text consequently.
  • Page 2 lines 78-79: the authors completely ignore the use of methods based on CTC physical characteristics. They should mention this additional approach and discuss the differences with the proposed one.
  • The authors consider that most of the current approaches select the cells on the basis of EpCAM expression, but this is partially true. They should compare their approach also to those methods that are not selecting the cells on this basis. See also previous comments on the same issue.
  • In the introduction the authors report a detailed summary of the results (Page 3, lines 130-144) while it is more appropriate to illustrate the aim of the study. Please modify consequently.
  • As regard to the preanalytical phase, the authors stated that the first negative selection-based-enrichment was performed within 24 hours from blood collection in EDTA tubes. Did the authors verify the stability of CTCs in the sample in the proposed time interval? Usually it is suggested to process CTC samples within few hours from collection since they are extremely fragile and submitted to lysis. The author shall report on this.
  • In paragraph 2.3 the staining methods it’s unclear to me. The method uses a dye for the nucleus, an APC conjugated anti EpCAm Ab and a secondary antibody (Alexa Fluor 488-conjugated donkey). What is the second antibody used for?
  • How can the author be sure that the CD45neg-EpCam neg cells are CTCs? Did they demonstrated that by using other tumor related markers? Can they exclude that these are all or mostly leucocytes?
  • I wonder if the use of a secondary Ab (to detect anti-EpCAM positive IgG- bound cells?) can increase the levels of unspecific binding. Can this be the reason why a high number of CD45neg-EpCAMneg cells have been identified in both control subjects and patients (see table 1)? How can the authors explain these huge number of CTCs negative for EpCAM in the case study?
  • If I correctly understand, No CD45 antibody was used for staining. The CD45 neg cells are defined on the basis of the fact that these cells are not depleted by using the Depletion kit, but in my opinion most of the CD45neg-EpCaM neg cells are leucocytes that have not been retained by the depletion method. Can you demonstrate that this is not true?
  • Did the authors compare their results with those obtained by filtration based methods or other methods based on the physical properties of the CTCs? They should provide evidences that the result of their counting is reliable and in line at least with methods not using EpCaM Ab for CTC capture.
  • In table 1 the value for the CEA levels were not reported for free of CRC subjects. Was CEA measured in the control population? CEA levels are often different from 0 also in normal subjects. Moreover, CEA varies also depending on age, thus the differences in mean age between controls and CRC patients can lead to unreliable results. How could the authors include CEA in the final evaluation of the results? How was the ROC curve obtained? Patients versus control subjects? For the control subjects CEA levels were assumed to be 0?
  • It is not clear which are the populations used to build the ROC curves. CRC patients versus controls? What about the CEA levels in the control population?
  • Referring to the combination in Figure 5, Controls have not been evaluated in comparison to patients. The authors shall provide these evaluation results.

Author Response

Reviewer 2

General Comments:

The manuscript deals with a hot-topic in the CTC field: EpCam positive CTCs versus EpCaM negatives ones.

The main problem in distinguishing these two population is analytical and derives from the sensitivity and the specificity of the used approach.

The proposed method uses just one EpCam antibody for positive selection, but does not use a CD45 fluorescently label antibody to verify the population obtained after depletion. It seems that the CD45neg/EpCaM neg cells are really very numerous to be all CTCs.

The authors should comment on this and compare their results with studies not using the recognition of biological markers for CTC selection like those base on the physic properties of the CTCs (filtration, dielectrophoresis, or other principles). Moreover, the comparison with the control population requires a matched group for age in which the values of CEA have been measured (CEA varies with the subject's age).

Although the question the authors aim to solve is interesting, the method used to achieve the goal should be carefully evaluated and correctly compared to an appropriate control group. Below you will find some of the major concerns relatively to the submitted manuscript.

Response: Reviewer’s insightful comments are appreciated. Regarding the fluorescent dye staining issue, we did use CD 45 fluorescent dye to label human CD45 on cells’ surface. In this study, Alexa Fluor 488-conjugated donkey anti-mouse IgG secondary antibody was used to label human CD45 marker, which could bind to the residual leukocytes coated with the commercial mouse anti-human CD45 antibodies (EasySep™ Human CD45 Depletion Cocktail II contained in EasySep Human CD45 Depletion Kit, StemCell Technologies, Vancouver, BC, Canada) after the CD45 depletion process. Hence, in following flow cytometry-based CTC enumeration, we fluorescently stained and labeled the human CD45 in the cells’ surface to avoid the leukocyte contamination or misjudgment issue. In order to make this clear, some of the descriptions have been newly supplemented in the manuscript. (Please refer to the manuscript of Highlighted Revision; Page 5).

Regarding the consideration on age, all analysis for performance of AUROC were applied by logistic regression, both sex and age have been considered as adjusted variables. The direct adjustment with multiple logistic regression was employed in our study due to sample size restriction, not sufficient samples for matching approach.

Specific comment 1:

The authors often refer to “Conventional CTC enumeration” apparently referring to CellSearch CTC counting. In my opinion would be better to specify the method they refer to, since there are other methods for CTC counting (e.g. those based on filtration) that don’t show this drawback.

Response: We thank the reviewer for reminding us this issue. The descriptions in the manuscript have now been improved as suggested by the reviewer

(Please refer to the manuscript of Highlighted Revision; Page 2, 3, 16).

Specific comment 2:

Page 2 line 70 and line 75: the authors refer to one report published 2014 (I guess the one from Cristofanilli), but they added 3 different, more recent references. Please modify references and/or text consequently.

Response: The mistake has now been corrected.

 (Please refer to the manuscript of Highlighted Revision; Page 2 & 17).

Specific comment 3:

Page 2 lines 78-79: the authors completely ignore the use of methods based on CTC physical characteristics. They should mention this additional approach and discuss the differences with the proposed one.

Response: The discussions relevant to the physical-based schemes for CTC isolation/ enumeration have now been supplemented in the manuscript

(Please refer to the manuscript of Highlighted Revision; Page 3).

Specific comment 4:

The authors consider that most of the current approaches select the cells on the basis of EpCAM expression, but this is partially true. They should compare their approach also to those methods that are not selecting the cells on this basis. See also previous comments on the same issue.

Response: We thank the reviewer’s insightful comments. The original descriptions and discussions relevant to the CTC enumeration approaches in the manuscript have now been improved

(Please refer to the manuscript of Highlighted Revision; Page 3).

Specific comment 5:

In the introduction the authors report a detailed summary of the results (Page 3, lines 130-144) while it is more appropriate to illustrate the aim of the study. Please modify consequently.

Response: Agreed. These sentences have been rewritten as- “Therefore, our study aimed to combine CD45neg EpCAMpos CTCs and CD45neg EpCAMneg cells to predict CRC, besides, based on existing conventional serum biomarker CEA, to evaluate add-on combining both CTC molecular biomarkers whether can improve the performance of AUROC for advanced CRC and CRC metastasis prediction.”

(Please refer to the manuscript of Highlighted Revision; Page 3-4).

Specific comment 6:

As regard to the preanalytical phase, the authors stated that the first negative selection-based-enrichment was performed within 24 hours from blood collection in EDTA tubes. Did the authors verify the stability of CTCs in the sample in the proposed time interval? Usually it is suggested to process CTC samples within few hours from collection since they are extremely fragile and submitted to lysis. The author shall report on this.

Response: We thank the reviewer for the suggestion. According to previous reports, an EDTA tube could protect CTCs from alteration of gene and protein expression for at least 72 hr under low temperatures (2~8 °C) condition [1]. This information has been newly supplemented in the manuscript.

(Please refer to the manuscript of Highlighted Revision; Page 4 & 19).

[1] Apostolou, P.; Ntanovasilis, D.A.; Papasotiriou, I. Evaluation of a simple method for storage of blood samples that enables isolation of circulating tumor cells 96 h after sample collection. Journal of Biological Research-Thessaloniki 2017, 24; DOI:ARTN 1110.1186/s40709-017-0068-9.

Specific comment 7:

In paragraph 2.3 the staining methods it’s unclear to me. The method uses a dye for the nucleus, an APC conjugated anti EpCAm Ab and a secondary antibody (Alexa Fluor 488-conjugated donkey). What is the second antibody used for?

Response: We are really sorry for the unclear descriptions on the staining methods. In this study, Alexa Fluor 488-conjugated donkey anti-mouse IgG secondary antibody was used to label human CD45 marker, which could bind to the residual leukocytes coated with the commercial mouse anti-human CD45 antibodies (EasySep™ Human CD45 Depletion Cocktail II contained in EasySep Human CD45 Depletion Kit, StemCell Technologies, Vancouver, BC, Canada) after the CD45 depletion process. The relevant descriptions in the manuscript have now been improved.

(Please refer to the manuscript of Highlighted Revision; Page 5).

Specific comment 8:

How can the author be sure that the CD45neg-EpCam neg cells are CTCs? Did they demonstrated that by using other tumor related markers? Can they exclude that these are all or mostly leucocytes?

Response: The reviewer’s insightful comments are appreciated. According to our previous study (PS: the description in the introduction section of this manuscript; page 3), first, we found that CD45neg EpCAMneg cell population might contain some cancer cells relevant to cancer metastasis due to their higher level of vimentin gene expression as compared to that of healthy blood donors. However, the real cell composition in the CD45neg EpCAMneg cell population have not yet been thoroughly understood, still being waiting for further studies. Moreover, as abovementioned, a commercial-available CD45 depletion kit and Alexa Fluor 488-conjugated donkey anti-mouse IgG secondary antibody were used in our study to exclude and distinguish the leukocytes in the CTC enumeration process. Therefore, the CD45neg EpCAMneg cell population had excluded the leukocytes.

Specific comment 9:

I wonder if the use of a secondary Ab (to detect anti-EpCAM positive IgG- bound cells?) can increase the levels of unspecific binding. Can this be the reason why a high number of CD45neg-EpCAMneg cells have been identified in both control subjects and patients (see table 1)? How can the authors explain these huge number of CTCs negative for EpCAM in the case study?

Response: We are sorry for the unclear description of Alexa Fluor 488-conjugated donkey anti-mouse IgG secondary antibody in this manuscript. As abovementioned in the previous response, Alexa Fluor 488-conjugated donkey anti-mouse IgG secondary antibody was used to label human CD45 marker. Hence, there is no significant unspecific binding issue between Alexa Fluor 488-conjugated donkey anti-mouse IgG secondary antibody and APC-conjugated rabbit anti-human EpCAM antibody.

As for the questions about the huge number of CD45neg EpCAMneg cells in healthy donors and patients, in fact, the numbers of CD45neg-EpCAMneg cells are not high and can be explained given the nature of flow cytometry. As described and illustrated in Fig. R1 in ref. 1, the existence of CD45neg-EpCAMneg cells that could be identified in both control subjects and patients has also been reported in the previous studies [1]. There was no statistical significance between these groups, which could be thought of as background or natural noise in the flow cytometry system. We believe that the background cells did not alter the main findings in this study.

Please see figure in attached file

Fig. R1. cited from [1]

Finally, considering the highly clinical application potential of CD45neg EpCAMneg cells shown in this study, the details of this cell population (e.g., their physiological role) are suggested to be further investigated. The above descriptions have now been supplemented in the manuscript. (Please refer to the manuscript of Highlighted Revision; Page 15).

[1] Bantikassegn A, Song X, Politi K. Isolation of epithelial, endothelial, and immune cells from lungs of transgenic mice with oncogene-induced lung adenocarcinomas. Am J Respir Cell Mol Biol 2015; 52: 409-417.

Specific comment 10:

If I correctly understand, No CD45 antibody was used for staining. The CD45 neg cells are defined on the basis of the fact that these cells are not depleted by using the Depletion kit, but in my opinion most of the CD45neg-EpCaM neg cells are leucocytes that have not been retained by the depletion method. Can you demonstrate that this is not true?

Response: We are really sorry for the unclear descriptions. As abovementioned in the previous response, Alexa Fluor 488-conjugated donkey anti-mouse IgG secondary antibody was used to label human CD45 marker, which could bind to the residual leukocytes coated with the commercial mouse anti-human CD45 antibodies (EasySep™ Human CD45 Depletion Cocktail II contained in EasySep Human CD45 Depletion Kit, StemCell Technologies, Vancouver, BC, Canada) after the CD45 depletion process. Hence, there is no significant leukocyte contamination or misjudgment issue in further flow cytometry-based CTC enumeration. The original descriptions have now been improved

(Please refer to the manuscript of Highlighted Revision; Page 5).

Specific comment 11:

Did the authors compare their results with those obtained by filtration based methods or other methods based on the physical properties of the CTCs? They should provide evidences that the result of their counting is reliable and in line at least with methods not using EpCaM Ab for CTC capture.

Response: This question from the reviewer is very important and practical. To answer that, we would like to mention a very similar comparison between the flow cytometry platform and CTC technology by physical properties [1]. In the literature, many investigators have attempted to answer the question. If we compare the CTC counts between flow cytometry protocol and ScreenCell CYTO (by size and deformability of cells) devices on the same clinical samples, the numbers obtained are quite similar, which suggests that CTC counting by flow cytometry is a repeatable measurement compared with other methods [1].

Please see figure in attached file

Fig R2. Cited from [1]. Correlation between ScreenCell CYTO device (CE IVD) and flow cytometry counts obtained using our staining and strategy. For each donor, the sample was treated in parallel with the ScreenCell technology (from 3 mL of blood) and our flow cytometry technique (from 1 mL of blood). Counts obtained by the 2 techniques are compared: on the y axis the count obtained with the flow cytometry technique, on the x axis the count obtained with ScreenCell technology for the same donor. Both counts strongly correlate, with a Spearman’s correlation R of 0.631 (P < 0.0001). Forty-three samples from donors with metastatic breast or colon cancers are represented in this graph. The correlation was established using the nonparametric Spearman’s test (2-tailed, confidence interval of 95%). A P value < 0.05 was considered statistically significant.

[1] Lopresti A, Malergue F, Bertucci F et al. Sensitive and easy screening for circulating tumor cells by flow cytometry. JCI Insight 2019; 5.

Specific comment 12:

In table 1 the value for the CEA levels were not reported for free of CRC subjects. Was CEA measured in the control population? CEA levels are often different from 0 also in normal subjects. Moreover, CEA varies also depending on age, thus the differences in mean age between controls and CRC patients can lead to unreliable results. How could the authors include CEA in the final evaluation of the results? How was the ROC curve obtained? Patients versus control subjects? For the control subjects CEA levels were assumed to be 0?

Response: According to the clinical scenario, CEA is not applied for normal subjects. Therefore, we did not include CEA level for CRC prediction, but applied for prediction on advanced or metastatic CRC. To clarify this point, first, we added one sentence in each section of Results part to state which subjects/patients were conducted for analysis- in Results 3.2 for CRC prediction, we added “There were 144 subjects (71 free of CRC subjects vs. 73 CRC patients) were conducted to evaluate the performance of CRC prediction.” (Please refer to the manuscript of Highlighted Revision; Page 8); in Results 3.4 for advanced/ metastatic CRC prediction, we added “There were 73 CRC patients only were included for advanced/ metastatic CRC pre-diction evaluation (non-advanced: advanced 27:46; non-metastatic: metastatic 59:14). (Please refer to the manuscript of Highlighted Revision; Page 10).

Second, we also added footnotes at the end of each Table to clarify which subjects were included. We added following:

Table 2: **: analysis included 71 free of CRC subjects vs. 73 CRC patients.

Table 3: **: analysis included 73 CRC patients only (non-advanced vs. advanced= 27 vs. 46).

Figure 3: (71 free of CRC subjects vs. 73 CRC patients)

Figure 5: (73 CRC patients only)

Figure 6: (non-advanced vs. advanced= 27 vs. 46).

Figure 7: (non-metastatic vs. metastatic= 59 vs. 14).

(Please refer to the manuscript of Highlighted Revision; Page 8-9, 11-14).

Specific comment 13:

It is not clear which are the populations used to build the ROC curves. CRC patients versus controls? What about the CEA levels in the control population?

Response: Same as Response for Specific comment 12.

Specific comment 14:

Referring to the combination in Figure 5, Controls have not been evaluated in comparison to patients. The authors shall provide these evaluation results.

Response: Same as Response for Specific comment 12. We have added the footnote for Figure 5. The add-on prediction could not achieve due to lack of CEA level for CRC free subjects. Furthermore, it is not fit the clinical practice scenario because the CEA is not applied for healthy subjects.

(Please refer to the manuscript of Highlighted Revision; Page 11).

Reviewer 3 Report

Review of scientific report entitled “Performance evaluation of combining multiple parameters, including the circulating tumor cell count, CD45neg EpCAMneg cell count or carcinoembryonic antigen level, for the prediction of colorectal cancer and its status” by Sherry Yueh-Hsia Chiu et al.

The authors aimed to explore whether any combination of Conventional circulating tumor cells (CTCs), CD45neg EpCAMneg cells and carcinoembryonic antigen (CEA) could improve the predictive performance for colorectal cancer (CRC) or its status. In this work, the authors quantified these two cell populations in healthy donors and CRC patients. Results revealed that enumeration of these two cell populations was able to discriminate healthy donors from CRC patients, even patients with non-advanced CRC. Moreover, combination of the two cell populations showed improved performance (AUROC: 0.893) for CRC prediction over the use of only one population. Compared with CEA alone, combination of the three biomarkers increased the performance (AUROC) for advanced CRC prediction from 0.643 to 0.727. Compared with that of CEA alone for metastatic CRC prediction, the AUROC was increased from 0.780 to 0.837 when the CTC count was included. Overall, this study demonstrated that combination of these two cellular biomarkers with CEA improved the predictive performance for CRC and its status.

Although the work showed interesting data, there are some major criticisms:

  1. Authors should also correlate their results with other clinical and laboratory parameters. For example, recent work has shown that there is a significant correlation between the number of circulating cancer cells and MSI (James W. T. Toh et al. Cells. 2020 Feb; 9: 425). Moreover, the authors should also correlate their parameters with mutational status of KRAS, NRAS and BRAF genes.
  2. The search and characterization of circulating tumor cells however requires the use of flow cytometers and various antibodies that could affect the analysis of this parameter. Authors should discuss these limitations. Would it be useful to combine a molecular parameter such as circulating DNA or mutation of RAS-BRAF genes also on circulating tumor cells?
  3. Authors showed that combination of the three biomarkers increased the performance (AUROC) for advanced CRC prediction from 0.643 to 0.727. Authors should also perform a multiple regression analysis of the three parameters to test their predictive ability alone or in combination.

Author Response

Reviewer 3

General Comments:

Review of scientific report entitled “Performance evaluation of combining multiple parameters, including the circulating tumor cell count, CD45neg EpCAMneg cell count or carcinoembryonic antigen level, for the prediction of colorectal cancer and its status” by Sherry Yueh-Hsia Chiu et al.

The authors aimed to explore whether any combination of Conventional circulating tumor cells (CTCs), CD45neg EpCAMneg cells and carcinoembryonic antigen (CEA) could improve the predictive performance for colorectal cancer (CRC) or its status. In this work, the authors quantified these two cell populations in healthy donors and CRC patients. Results revealed that enumeration of these two cell populations was able to discriminate healthy donors from CRC patients, even patients with non-advanced CRC. Moreover, combination of the two cell populations showed improved performance (AUROC: 0.893) for CRC prediction over the use of only one population. Compared with CEA alone, combination of the three biomarkers increased the performance (AUROC) for advanced CRC prediction from 0.643 to 0.727. Compared with that of CEA alone for metastatic CRC prediction, the AUROC was increased from 0.780 to 0.837 when the CTC count was included. Overall, this study demonstrated that combination of these two cellular biomarkers with CEA improved the predictive performance for CRC and its status.

Although the work showed interesting data, there are some major criticisms:

Specific comment 1:

Authors should also correlate their results with other clinical and laboratory parameters. For example, recent work has shown that there is a significant correlation between the number of circulating cancer cells and MSI (James W. T. Toh et al. Cells. 2020 Feb; 9: 425). Moreover, the authors should also correlate their parameters with mutational status of KRAS, NRAS and BRAF genes.

Response: Thanks for suggestion. However, we did not have those biomarkers so far, but this is good research topic for next step. Therefore, we added some paragraphs to discussion this point. We added:

“The majority of CRC patients come from sporadic cases, around 80%, others are partially derived from family history and hereditary susceptibility interacting with environment factors. The CRC incidence rate is highly correlated with age; for those who tend to have high risk of sporadic CRCs, they are high likelihood of being late-onset cases, but with high probability of being MSI (microsatellite instability), MLH1 methylated, BRAF, and KARS mutation [1]. Some studies also demonstrated the mutation associated with advanced and metastasis CRCs [2]. Based on the meta-analysis evidence, these biomarkers also play important roles for clinical treatment of targeted therapy designed by epidermal growth factor receptor (EGFR) signaling [3]. Therefore, the biomarkers examination is emerging from biotech development, especially on the isolation methods from blood samples [4, 5] rather than from tumor tissues, which is convenient for clinical practice using CTCs detection [6]. Based on the development of CTCs examination and knowledge on the specific biomarkers for CRC targeted therapy application, it is promising to combine those together toward the precision medicine and healthcare application for CRC prognosis prediction. Further add-on molecular markers, such as circulating DNA or mutation of RAS-BRAF genes on CTCs, are highly warranted.

Recently, Toh et al. reported those CRC patients with high level of MSI g CTCs are tend to increase CTCs. Furthermore, compared the different time-point with pre-, intra-, post-operation for CRC, those CRCs with high MSI were significantly associated with increasing CTCs level during intra- and post- operative time points [7], which indicated the CTCs might be influenced by clinical intervention, i.e. chemotherapy, surgery, or radiotherapy, etc. Therefore, the multiple time points with repeated measures for CTCs with longitudinal follow-up would need for future application.”

 (Please refer to the manuscript of Highlighted Revision; Page 15-16, 19).

[1] Alvarez, K., Cassana, A., De La Fuente, M., Canales, T., Abedrapo, M., & López-Köstner, F. (2021). Clinical, Pathological and Molecular Characteristics of Chilean Patients with Early-, Intermediate- and Late-Onset Colorectal Cancer. Cells, 10(3), 631. https://doi.org/10.3390/cells10030631  

[2] László, L., Kurilla, A., Takács, T., Kudlik, G., Koprivanacz, K., Buday, L., & Vas, V. (2021). Recent Updates on the Significance of KRAS Mutations in Colorectal Cancer Biology. Cells, 10(3), 667. https://doi.org/10.3390/cells10030667

[3] Therkildsen, C., Bergmann, T. K., Henrichsen-Schnack, T., Ladelund, S., & Nilbert, M. (2014). The predictive value of KRAS, NRAS, BRAF, PIK3CA and PTEN for anti-EGFR treatment in metastatic colorectal cancer: A systematic review and meta-analysis. Acta oncologica (Stockholm, Sweden), 53(7), 852–864. https://doi.org/10.3109/0284186X.2014.895036

[4] Mohamed Suhaimi, N. A., Foong, Y. M., Lee, D. Y., Phyo, W. M., Cima, I., Lee, E. X., Goh, W. L., Lim, W. Y., Chia, K. S., Kong, S. L., Gong, M., Lim, B., Hillmer, A. M., Koh, P. K., Ying, J. Y., & Tan, M. H. (2015). Non-invasive sensitive detection of KRAS and BRAF mutation in circulating tumor cells of colorectal cancer patients. Molecular oncology, 9(4), 850–860. https://doi.org/10.1016/j.molonc.2014.12.011

[5] Kondo, Y., Hayashi, K., Kawakami, K., Miwa, Y., Hayashi, H., & Yamamoto, M. (2017). KRAS mutation analysis of single circulating tumor cells from patients with metastatic colorectal cancer. BMC cancer, 17(1), 311. https://doi.org/10.1186/s12885-017-3305-6

[6] Diao, Z., Han, Y., Chen, Y., Zhang, R., & Li, J. (2021). The clinical utility of microsatellite instability in colorectal cancer. Critical reviews in oncology/hematology, 157, 103171. https://doi.org/10.1016/j.critrevonc.2020.103171

[7] Toh, J.W.T.; Lim, S.H.; MacKenzie, S.; de Souza, P.; Bokey, L.; Chapuis, P.; Spring, K.J. Association between Microsatellite Instability Status and Peri-Operative Release of Circulating Tumour Cells in Colorectal Cancer. Cells 2020, 9, 425. https://doi.org/10.3390/cells9020425

Specific comment 2:

The search and characterization of circulating tumor cells however requires the use of flow cytometers and various antibodies that could affect the analysis of this parameter. Authors should discuss these limitations. Would it be useful to combine a molecular parameter such as circulating DNA or mutation of RAS-BRAF genes also on circulating tumor cells?

Response: The reviewer’s insightful comments are appreciated. This method of CTC counting has limitations of (1) background cell contamination when CTC is extremely rare; (2) incomplete or oversaturated antibody staining resulting in underestimated or overestimated cell counting. Fortunately, these issues could be mostly prevented by non-staining, isotype negative controls, and positive controls using cancer cell lines along with each CTC testing. The above descriptions have now been supplemented in the manuscript. (Please refer to the manuscript of Highlighted Revision; Page 14).

Besides, “Further add-on molecular markers, such as circulating DNA or mutation of RAS-BRAF genes on CTCs are highly warranted.” has now been supplemented in the manuscript. (Please refer to the manuscript of Highlighted Revision; Page 15). Meanwhile, the concept has been proven in some preliminary studies [1].

Please see figure in attached file

Fig R3. Cited from [1]. Number of CTCs during treatment and clinical outcome response in patient MMbraf5.

[1] Okuyama R, Kiniwa Y, Nakamura K et al. Usefulness of Monitoring Circulating Tumor Cells as a Therapeutic Biomarker in Melanoma with BRAF Mutation. 2020.

Specific comment 3:

Authors showed that combination of the three biomarkers increased the performance (AUROC) for advanced CRC prediction from 0.643 to 0.727. Authors should also perform a multiple regression analysis of the three parameters to test their predictive ability alone or in combination.

Response: Based on multiple logistic regressions, these results have been done and shown on Table 3 and Table 4 for advanced and metastatic CRC prediction, respectively, including one alone (1-type), 2-type, and 3-type combination. (see Table 3 and Table 4)

(Please refer to the manuscript of Highlighted Revision; Page 12-13).

Round 2

Reviewer 2 Report

Revision Manuscript cancers-1169819

Second round revision

I thank the author for the extensive improvements of the material and methods section that allowed a deep inside on the details of the method used.

They also added the cited papers from Cristofanilli in the appropriate position.

Thanks to the additional explanations, I hopefully understood the workflow of the assay, nevertheless I have still some concerns about the analytical and pre-analytical phase.

Regarding the pre-analytical phase, my own experience on CTCs storage does not confirm the findings of the cited paper on the stability of CTCs up to 72 hours at 4°C.

In my hands, the morphology of the cells appears to be damaged after only 3 hours at room temperature. I don’t think that 4°C can improve the time much longer…

The paper that was cited to claim CTC stability up to 72 hours provides data on the stability of some transcripts (most of them not found exclusively on CTCs like 18S) that is not sufficient to demonstrate the CTC stability themselves.

Regarding the spiked-in samples they start from about 750.000 cancer cells/ tube that is not a number suitable to mimic the patient’s conditions; among them some will probably show a normal morphology.  

Thus, I am strongly convinced that the storage conditions applied in the study might have introduced bias especially related to the ability of the antibody to recognize the epithelial marker.

The authors can find guidelines for CTC pre-analytical phase in some CTC documents developed recently by CEN (Technical Specifications CEN/TS 17390-1, CEN/TS 17390-2 and CEN/TS 17390-3 describe special measures that need to be taken to obtain good quality, high pure samples for human DNA and RNA isolated from CTCs examination and for CTC staining).

This issue can compromise the whole study results.  Thus a demonstration of the stability of the CTCs in the authors’ working conditions is of primary importance in respect to all the other problems.

Reviewer 3 Report

The revisions have enriched the manuscript that has reached priority for publication in Cancers.